# AdaptNet: Adaptive Learning from Partially Labeled Data for Abdomen Multi-Organ and Tumor Segmentation

JiChao Luo[1,2], Zhihong Chen[1,2] , Wenbin Liu[1], Zaiyi Liu[2,4], Bingjiang Qiu[2,3,4] , and Gang Fang[1]

[1] Institute of Computing Science and Technology, Guangzhou University, Guangzhou, 510006, China
[2] Department of Radiology, Guangdong Provincial People's Hospital (Guangdong Academy of Medical Sciences), Southern Medical University, Guangzhou 510080, China
[3] Guangdong Cardiovascular Institute, Guangdong Provincial People's Hospital, Guangdong Academy of Sciences, Guangzhou 510080, China
[4] Guangdong Provincial Key Laboratory of Artificial Intelligence in Medical Image Analysis and Application, Guangzhou 510080, China
qiubingjiang@gdph.org.cn, gangf@gzhu.edu.cn

**Abstract.** Due to the high costs associated with the labor and expertise required for annotating 3D medical images at the voxel level, most public and in-house datasets only include annotations of a single (or a few) organ or tumor. This limitation results in what is commonly referred to as the 'partial labeling/annotation problem'. In order to tackle this issue, we introduce an adaptive learning network, AdaptNet, to effectively segment multiple organs and tumors within partially labeled data from abdomen CT images. AdaptNet comprises three key components: a segmentation network, a pseudo-label generation network, and an adaptive controller responsible for generating dynamic weights. AdaptNet generates adaptive weights dynamically through the controller, which takes into account the balance of the partial labels and the corresponding pseudo-labels. This approach enables AdaptNet to efficiently and flexibly learn multiple organ and tumor information from the partial labeling/annotation dataset, which is typically performed by multiple or multi-head networks. We conduct validation on a large-scale partially annotated dataset under MICCAI FLARE 2023 challenge and demonstrate that the proposed AdaptNet outperforms the baseline method across the 13 different organ and tumor segmentation tasks. Our method achieves a mean organ Dice Similarity Coefficient (DSC) of 89.61% and a Normalized Surface Dice (NSD) of 94.94%, and a tumor DSC and NSD of 39.16% and 30.52% on the FLARE 2023 online validation. Additionally, in the Final Testing dataset, our method achieves a mean organ DSC and NSD of 89.34% and 95.26% and a tumor DSC and NSD of 54.59% and 40.78%, and the area under GPU memory-time curve is 33.35s and 84276 MB. The code is available at https://github.com/Prech-start/FLARE23_AdaptNet.

**Keywords:** Adaptive learning · partial labeling/annotation · Abdomen organ segmentation.

## 1   Introduction

Abdominal organs are quite common cancer sites, such as colon and rectum cancer, and pancreas cancer, which are the 2nd and 3rd most common cause of cancer death [21]. Computed tomography (CT) provides doctors with valuable prognostic information. During the diagnosis process, the doctor evaluates the lesion or organ by manual annotation in two dimensions plane on the CT, which leads to a tedious procedure in the clinical practice. Moreover, the structural complexity of abdominal organs and their cancers make the annotation process challenging [13]. Currently, there are many high-quality publicly available tumor datasets, such as liver cancer segmentation [3], lung nodule segmentation [1], etc. However, they are all for one single type of tumor. In terms of organ segmentation, several multi-organ segmentation datasets with all the organ labels have been released, e.g., BTCV [15], AMOS[12], etc. However, this kind of dataset with all organs or tumors annotated is almost unachievable in real clinical workflow. Utilizing these datasets inevitably creates partial labeling/annotation problems. Furthermore, there is still no general and publicly available dataset with 'partial labeling/annotation problems' for universal abdominal organs and pan-cancer segmentation nowadays. FLARE2023 challenge, an extension of FLARE2021 and FLARE2022 challenges, provides such an opportunity, which aims to promote the development of universal organ and tumor segmentation in abdominal CT scans. FLARE2023 showcases a rich variety of tumor types and a combination of multiple different organ annotations, as shown in Fig. 1. This imbalanced labeling could potentially lead to the failure of the segmentation methods.

Formerly, researchers have proposed some traditional segmentation methods gray-level based methods [14], Live wire segmentation approaches [23], and mathematical fitting procedure [5] for segmentation tasks which are more efficiently than manual segmentation methods. However, the traditional methods need manual design features. Compared with the traditional methods, Deep Learning (DL) methods demonstrate enhanced accuracy and much better generalization capacity. In recent years, regardless of various works based on fully supervision learning method [16] achieve State-of-the-Art (SOTA) performance in single data centers, many of which are small and single data center [20]. Furthermore, most of the SOTA methods cannot be easily verified and generalized in other datasets with imbalanced annotations.

To address the problem, this study intends to use the core concept of semi-supervised learning to effectively use unlabeled organ samples to improve model performance. Semi-supervised learning potentially learns wrong information from incorrect pseudo-labels, which would lead to performance degradation. Normally, selecting high-confidence predictions can fix the problem of performance degradation. However, this way would exclude a large amount of unlabeled data from

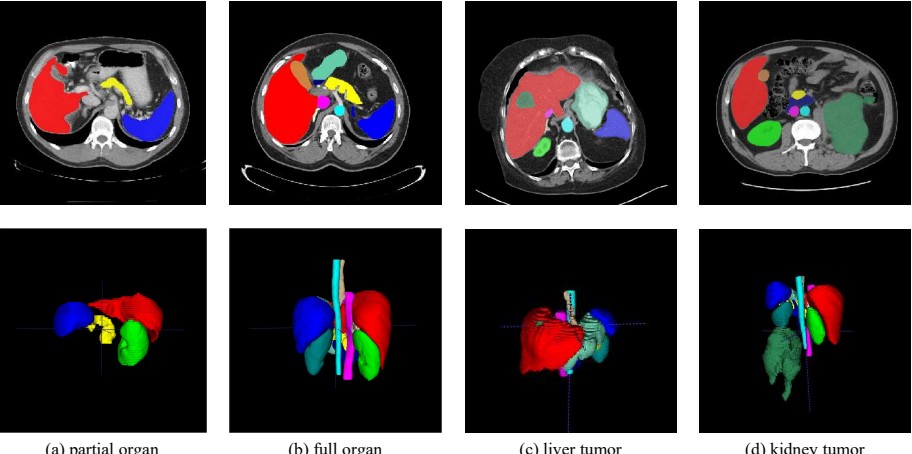

(a) partial organ      (b) full organ      (c) liver tumor      (d) kidney tumor

**Fig. 1.** Samples of imbalanced annotations in FLARE2023. Partial organ annotations are observed in some cases, as depicted in case (a). In other cases, all organs are annotated, but tumors are absent, as shown in case (b). There are also cases with annotations covering both tumors and organs, as demonstrated in cases (c) and (d).

the training process, resulting in insufficient model training. Furthermore, this way leads to the low-quality pseudo-labels not being utilized in training. Therefore, based on that, we propose an adaptive learning segmentation method to efficiently utilize and learn pseudo-labels.

In this paper, we propose an automatic segmentation method, AdaptNet, for abdominal organs and cancers based on FLARE2023 dataset with imbalanced partial labeling. The proposed framework AdaptNet mainly contains three components: a pseudo-label generation network that creates the class-wise annotations which not exist in the true labels, a controller responsible for generating dynamic weights, and a segmentation network that segments lesions and organs based on adaptive weights generated by dynamic weights controller. The main contributions of this work are summarized as follows: (1) Through the proposed AdaptNet, pseudo-labels have been effectively utilized and learned, which introduces the unlabeled organ information, while also avoiding the misleading from incorrect pseudo-labels. (2) To balance the pseudo-labels and the original label, dynamic weights are generated automatically by a controller. (3) To mitigate the misleading of incorrect pseudo-labels, an adaptive loss approach is employed to train the segmentation model. Experiments show the effectiveness of the proposed AdaptNet for the partial labeling problem.

## 2   Method

### 2.1   Preprocessing

**Resample and normalization** We resample the pixel spacing to (2.2838, 1.8709, 1.8709) for all cases, and clip the pixel value based on the Hounsfield units to $[-160, 240]$, and normalize all the cases in $[0, 1]$ to ensure data stability and consistency.

**Cropping the data** To reduce redundant or irrelevant information and save computing resources, all the original CT matrix is cropped according to the foreground markers generated by original labels and pseudo-labels (the details are in the next section).

**Data augmentation** In order to prevent the model from over-fitting, data augmentation is used in this study. The augmentation approaches of nnU-Net methodology [11] have been utilized.

### 2.2   Proposed Method

Specifically, the proposed AdaptNet contains a pseudo-label generator network which is followed by a label filling module, a baseline network which is to make a segmentation prediction, and a dynamic weights controller which is mainly made up of an adaptive weight calculation (AWC) module, as shown in Fig. 2.

**Pseudo-Label Generator** Pseudo-labels contain valuable information about the location and boundary of target organs and tumors during training, which enhances the model's discriminative ability. By incorporating pseudo-labels, pseudo-label generator arguments the datasets and promotes the model to learn more boundary information from unlabeled organs in true labels. Here, we apply the segmentation network from [10] as the pseudo-label generator network, which has achieved remarkable results in the FLARE2022 challenge.

**Segmentation Network** The baseline network is built upon nnU-Net [11], utilizing parameters generated using the nnU-Net methodology.

**Label Filling Module** To incorporate pseudo-labels into the true labels, Label Filling Module is used after pseudo-labels were generated. The details are illustrated in Fig. 2. In general, the Label Filling Module can be expressed in the following equation:

$$ML = R(PL, U_{TL} \cap U_{PL}, 0) + TL \tag{1}$$

where the $U_{TL}$ and $U_{PL}$ denote the list of classes for true label (TL) and pseudo-label (PL). The expression $R(S, I, 0)$ signifies the substitution of the intersection $I$ with the value 0 within the set $S$, and then combining TL and PL to form a mixed-label (ML).

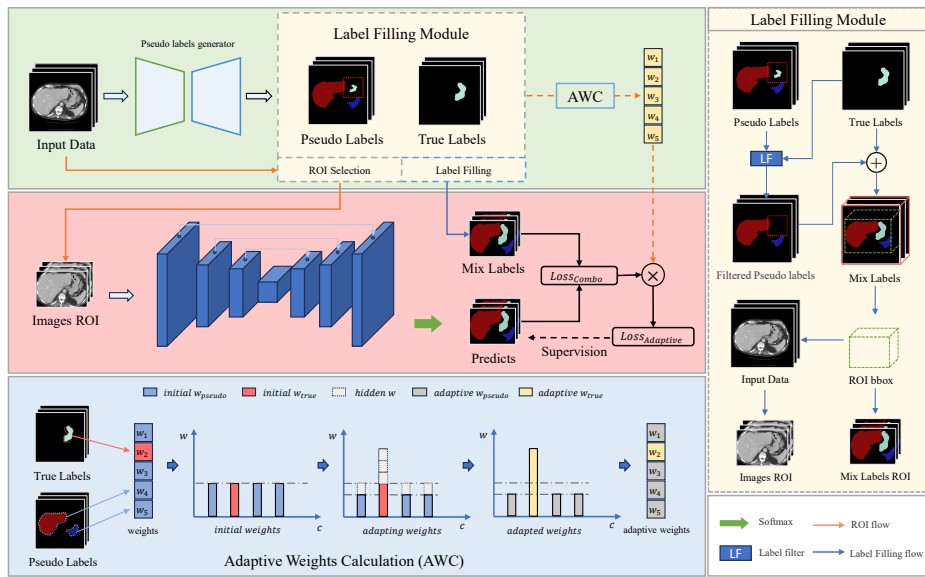

**Fig. 2.** Overview of our proposed AdaptNet. Green block: Generate the mix label and ROI bounding box by Label Filling Module, and weight of adaptive loss calculation by AWC. Pink block: Baseline segmentation network. Blue block: Adaptive weight calculation (AWC) module: calculate the weight according to the unique object class between the pseudo-label and the true label. Yellow Block: Label Filling Module filters the interfering information and combines the pseudo-label and true label into a mix-label.

**Adaptive Weight Calculation** Considering the potentially misleading effects of pseudo-labels during training, we introduce an adaptive loss to impose constraints. The main idea of adaptive loss is to automatically weaken learning efficiency from pseudo-labels while amplifying the guiding capability of true labels during training. For the purpose of this paper, we define symbol $C_O$ as $[1, class\_count]$, where the $class\_count$ means the total class counts for the segmentation task, and suppose $C_{pseudo}$ represents the unique indexes collection of pseudo-label. Its definition is

$$C_{pseudo} = \{c_i \mid c_i \in C_O, c_i \text{ is pseudo label}\},$$

where $i$ is the $i$-th class index. Then, indexes collection of true label $C_{true}$ is

$$C_{true} = \{c_i \mid c_i \in C_O, c_i \text{ is true label}\}.$$

Then, the updated steps of loss weight for each target class are as follows:

$$w_i^o = \frac{1}{class\_count}, i \in C_O, \tag{2}$$

$$w_i^p = \xi_{ada} * w_i^o, i \in C_{pseudo}, \tag{3}$$

$$w_i^t = w_i^o + \frac{\sum^{c \in C_{pseudo}}(w_c^o - w_c^p)}{\mid C_{true} \mid}, i \in C_{true}, \tag{4}$$

where $w_i^p$ and $w_i^t$ represent the weight of the $i$-th class in pseudo-label and true label, respectively. The $\xi_{ada}$ is an adjustable parameter to control attention to the true label. It is initialized to a default value of 0.5. The $|C_{true}|$ equals with class number in true label.

In general, the weight $w_i$ of the $i$-th class can be defined as follows:

$$w_i = \begin{cases} w_i^p, & i \in C_{pseudo} \\ w_i^t, & i \in C_{true} \end{cases} \tag{5}$$

In this way, the model can pay more attention to the organ with real labels and also learn the shape or location information of unlabeled organs via their corresponding pseudo-labels. In other words, the true label gains a dynamic higher loss score than pseudo-labels according to the label status of each patch. Therefore, the Adaptive Weight module suppresses gradients generated by features in the filled labels that could disrupt training and enhance the learning capacity for the true annotations.

Then, we combine the adaptive weight and $ComboLoss$ function which is combined with $DiceLoss$ and $CELoss$. The $ComboLoss$ converges considerably faster than cross-entropy loss during training[25]. It is defined as:

$$L_{CE}(y, \hat{y}, w) = \sum_i^{C_O} w_i(-\frac{1}{N}\sum_{j=1}^N y_j^i log(\hat{y}_j^i) + (1 - y_j^i)log(1 - \hat{y}_j^i)), \tag{6}$$

$$L_{Dice}(y, \hat{y}, w) = \sum_{i}^{C_O} w_i(1 - \frac{2 \sum_{j=1}^{N} y_j^i \hat{y}_j^i}{\sum_{j=1}^{N} y_j^i + \hat{y}_j^i}), \qquad (7)$$

$$loss(y, \hat{y}, w) = \alpha_{ce} * L_{CE}(y, \hat{y}, w) + \alpha_{dc} * L_{Dice}(y, \hat{y}, w), \qquad (8)$$

where the $y_j^i$ and $\hat{y}_j^i$ mean the ground truth and the predicted probability of pixel $j$, respectively, and $N$ is the number of pixels. $\alpha_{ce}$ and $\alpha_{dc}$ are the hyperparameters to balance the contribution of $DiceLoss$ and $CELoss$. $\alpha_{ce}$ and $\alpha_{dc}$ are set to 0.5 in this study.

**Training Strategies** One of the obstacles to training 3D networks is the problem of "insufficient memory". A common solution is to train a 3D network from smaller sub-volumes (3D patches) and test it by sliding window. We set the step of the sliding window and use multithreaded preprocessing of CT image to reduce our inference time. The shape of the sliding window is consistent with the patch as shown in Table 3. Here, to reduce the inference time, the length of the step is $[5/6, 7/8, 9/10]$ times the window width for each axis instead of the default parameter $[1/2, 1/2, 1/2]$ of nnU-Net. Consequently, the inference time significantly decreases, e.g., from 72s to 48s for case 0048 in the environment of this study.

### 2.3  Post-processing

In the post-processing stage, we employ a connected component-based method after the segmentation prediction. Particularly in organ image segmentation, it helps remove the disconnected voxels, consequently, reducing false positives. In the study, the largest connected component of each segmented organ volume is simply selected.

## 3  Experiments

### 3.1  Dataset and evaluation measures

The FLARE 2023 challenge is an extension of the FLARE 2021-2022 [18][19], aiming to promote the development of foundation models in abdominal disease analysis. The segmentation targets cover 13 organs (liver, spleen, pancreas, right kidney (RK), left kidney (LK), stomach, gallbladder, esophagus, aorta, inferior vena cava (IVC), right adrenal gland (RAG), left adrenal gland (LAG), and duodenum) and various abdominal lesions, which cover various abdominal cancer types, such as liver cancer, kidney cancer, pancreas cancer, colon cancer, gastric cancer, and so on. The organ annotation process used ITK-SNAP [27], nnU-Net [11], and MedSAM [17]. The training dataset is curated from more than 30 medical centers under the license permission, including TCIA [4], LiTS [2], MSD [24], KiTS [8,9], autoPET [7,6], TotalSegmentator [26], and AbdomenCT-1K [20]. The training set includes 4000 abdomen CT scans where 2200 CT scans

with partial labels and 1800 CT scans without labels. The validation and testing sets include 100 and 400 CT scans. In this study, unlabeled images were not used. Only 2200 scans with partial labels have been used due to the computational resource limitation, and the 1800 unlabled images are not used. The frequency statistics about the 2200 cases regarding organ and tumor annotations are provided in Table 1. 5-fold cross-validation has been performed, in which 1760 cases are chosen as the training dataset, and the rest 440 cases are as the internal validation dataset in each fold.

**Table 1.** Organ annotation occurrence frequency(%) summary

| Target | Liver | RK | Spleen | Pancreas | Aorta | IVC | RAG |
|---|---|---|---|---|---|---|---|
| Frequency | 59.6 | 59.1 | 59.4 | 59.6 | 11.3 | 11.3 | 11.3 |
| Target | LAG | Gallbladder | Esophagus | Stomach | Duodenum | LK | Tumor |
| Frequency | 11.2 | 10.2 | 11.3 | 11.3 | 11.3 | 59.0 | 68.0 |

The evaluation metrics encompass two accuracy measures—Dice Similarity Coefficient (DSC) and Normalized Surface Dice (NSD)—alongside two efficiency measures—running time and area under the GPU memory-time curve. These metrics collectively contribute to the ranking computation. Furthermore, the running time and GPU memory consumption are considered within tolerances of 15 seconds and 4 GB, respectively.

### 3.2   Implementation details

**Environment settings** The development environments and requirements are presented in Table 2.

**Table 2.** Development environments and requirements.

| | |
|---|---|
| System | Ubuntu 23.04 |
| CPU | Intel(R) Core(TM) i9-10900X CPU@3.70GHz |
| RAM | 4×32GB; 2933MT/s |
| GPU | NVIDIA GeForce RTX™3090 24G |
| CUDA version | 12.0 |
| Programming language | Python 3.9.16 |
| Deep learning framework | Pytorch (Torch 2.0.1) |
| Code | https://github.com/Prech-start/FLARE23_AdaptNet |

**Training protocols** During the training phase, we set the batch size to 2 and randomly select all samples within each epoch. For each sample, we perform

random patch cropping with patch sizes of $(96, 128, 160)$. As for the optimizer, we utilize AdamW with a learning rate of 1e-3 and a weight decay of 1e-5. The learning rate updating follows the default mechanism of AdamW. Additional details are presented in Table 3.

**Table 3.** Training protocols.

| | |
|---|---|
| Network initialization | "he" normal initialization |
| Batch size | 2 |
| Patch size | 96×128×160 |
| Total epochs | 120 |
| Optimizer | AdamW with weight decay($\mu = 1e - 5$) |
| Initial learning rate (lr) | 0.001 |
| Lr decay schedule | halved by 200 epochs |
| Training time | 11 hours per fold |
| Loss function | Adaptive Loss |
| Number of model parameters | 30.8M[5] |
| Number of flops | 838.6116 G[6] |
| $CO_2$eq | 3.91908 Kg[7] |

## 4   Results and discussion

The best fold was selected via the results in the Public validation, as shown in Table 4. The result of Public Validation is calculated with the 50 open cases from 100 Validation set. The result for Online Validation is collected from FLARE2023 website. It is worth noting that in the metrics for public validation, we have included the standard deviation, represented as evaluation score±std. The std of the online validation is not available since it is not reported online. The results for the validation are listed in Table 5.

**Table 4.** Segmentation DSC(%) of five fold from Public Validation.

| Target | baseline | | label filling | | proposed | |
|---|---|---|---|---|---|---|
| | Organ | Tumor | Organ | Tumor | Organ | Tumor |
| fold0 | 34.84 | 36.89 | 89.25 | 40.94 | 88.96 | 45.12 |
| fold1 | 36.18 | 37.20 | 89.30 | 40.75 | 88.97 | 43.24 |
| fold2 | 35.79 | 34.15 | 89.05 | 42.02 | 89.02 | 45.35 |
| fold3 | 35.57 | 40.17 | 89.20 | 44.19 | 89.09 | 43.75 |
| fold4 | 35.07 | 39.61 | 89.22 | 41.73 | 88.94 | 45.04 |
| mean | 35.49 | 37.60 | 89.20 | 41.92 | 88.96 | 44.50 |

**Table 5.** Result in Public Validation, Online Validation and Final Testing.

| Target | Public Validation | | Online Validation | | Testing | |
|---|---|---|---|---|---|---|
| | DSC(%) | NSD(%) | DSC(%) | NSD(%) | DSC(%) | NSD (%) |
| Liver | 97.74±0.44 | 99.28±0.73 | 97.60 | 99.07 | 96.95 | 98.27 |
| RK | 94.44±7.76 | 95.92±8.61 | 93.83 | 95.36 | 93.73 | 94.75 |
| Spleen | 96.88±0.94 | 99.12±1.75 | 96.94 | 99.19 | 96.48 | 98.94 |
| Pancreas | 86.06±5.58 | 97.06±4.01 | 84.70 | 96.18 | 88.64 | 97.01 |
| Aorta | 94.74±1.25 | 98.78±2.29 | 94.74 | 98.72 | 94.97 | 99.53 |
| IVC | 88.62±7.60 | 91.26±7.90 | 88.30 | 90.60 | 88.83 | 92.09 |
| RAG | 81.41±12.23 | 94.97±13.68 | 81.43 | 95.51 | 81.65 | 95.30 |
| LAG | 82.64±5.66 | 95.96±4.34 | 80.86 | 94.37 | 81.94 | 94.79 |
| Gallbladder | 86.53±18.94 | 88.38±20.43 | 84.11 | 85.80 | 83.38 | 86.32 |
| Esophagus | 79.95±16.67 | 90.77±16.92 | 81.14 | 92.53 | 86.56 | 96.88 |
| Stomach | 93.14±3.20 | 97.25±4.10 | 93.67 | 97.59 | 93.53 | 97.17 |
| Duodenum | 81.51±7.90 | 94.80±5.91 | 81.43 | 94.54 | 84.02 | 94.27 |
| LK | 93.45±6.72 | 94.73±8.90 | 93.18 | 94.81 | 92.90 | 94.40 |
| Organ Average | 89.01±6.13 | 95.25±3.23 | 88.61 | 94.94 | 89.34 | 95.26 |
| Tumor | 43.75±35.21 | 35.46±29.93 | 39.16 | 30.52 | 54.59 | 40.78 |

### 4.1   Quantitative results on validation set

As shown in Table 6, the quantitative experiments have been carried out for more comprehensive ablation studies on the Pseudo-label filling and Adapting weight calculation. For the tumor segmentation, the proposed method performs better than the baseline model and the label-filling-based model, with an improvement of at least 0.0258 and 0.0295 in DSC and NSD scores, respectively. For the organ segmentation, the segmentation result of our proposed method is slightly worse (with a decline of only 0.0027 in DSC score) than the model that used pseudo-label filling. Specifically, comparisons with Quantitative evaluation in Table 6 and annotation statistics in Table 1 illustrate that the baseline model is invalid in segmenting the organs with low frequency, i.e., aorta (0.113), IVC (0.113), RAG (0.113), LAG (0.112), gallbladder (0.102), esophagus (0.113), stomach (0.113), and duodenum (0.113). The model's ability is strengthened in tumors and organs with high frequency (e.g., liver, spleen, etc. ). It also demonstrates the effectiveness of pseudo-label filling in the segmentation task with imbalance annotations. The proposed AdaptNet approach improves segmentation of the part of objects with high frequency (i.e., RK (0.591), LK (0.590), and tumor (0.680)), while the segmentation results from AdaptNet are not as promising as the model used pseudo-label filling for the organs with low frequency. According to the weight calculation algorithm and the frequency of organ annotation occurrence, it can be inferred that this situation is reasonable in that the lower the labeling frequency, the less guided by the real annotation.

**Table 6.** Overview of Ablation Experiment Results. Note: Label filling: baseline + Label filling module. Proposed: baseline + Label filling module + Adaptive weight calculation.

| Target | Baseline | | Label filling | | Proposed | |
|---|---|---|---|---|---|---|
| | DSC(%) | NSD(%) | DSC(%) | NSD(%) | DSC(%) | NSD(%) |
| Liver | 90.78 | 91.85 | 97.76 | 99.26 | 97.72 | 99.22 |
| RK | 89.08 | 90.35 | 93.84 | 95.25 | 94.02 | 95.39 |
| Spleen | 91.71 | 93.57 | 96.90 | 99.22 | 96.85 | 99.10 |
| Pancreas | 80.25 | 91.39 | 85.87 | 96.93 | 85.85 | 96.90 |
| Aorta | 1.73 | 1.59 | 94.78 | 98.90 | 94.68 | 98.69 |
| IVC | 2.29 | 2.12 | 89.32 | 92.12 | 88.62 | 91.25 |
| RAG | 6.96 | 7.61 | 81.24 | 94.86 | 81.26 | 94.87 |
| LAG | 6.23 | 7.26 | 82.33 | 95.73 | 81.85 | 95.35 |
| Gallbladder | 10.24 | 10.03 | 85.08 | 87.02 | 83.96 | 85.74 |
| Esophagus | 3.78 | 4.50 | 80.54 | 91.37 | 80.25 | 91.13 |
| Stomach | 3.54 | 3.85 | 93.80 | 97.70 | 93.40 | 97.69 |
| Duodenum | 0.46 | 0.59 | 82.06 | 94.88 | 81.31 | 94.68 |
| LK | 88.64 | 89.76 | 93.15 | 94.19 | 93.37 | 94.59 |
| Organ Average | 36.60 | 38.04 | 88.97 | 95.19 | 88.70 | 94.97 |
| Tumor | 37.60 | 27.68 | 41.93 | 32.34 | 44.51 | 35.29 |

## 4.2   Qualitative results on validation set

In this section, we show the two good segmentation cases and two bad segmentation cases.

**Good segmentation cases** As shown in case-0087 of Fig. 3, the baseline method is not available to segment the IVC, aorta, stomach, duodenum and RAG. Meanwhile, the baseline method misclassifies part of LK as spleen. The label filling method can only segment part of the duodenum. Compared to the under-segmentation of the baseline method and the label filling method in the kidney, our method performs much better in the tumor. In case-0057 of Fig. 3, the tumor in RK, stomach, aorta, LK and IVC are not segmented in the baseline method. The part of LK is misclassified as part of the tumor and the lesion in LK is under-segmentation by the label filling method, while the proposed AdaptNet can almost segment the tumor in LK, however, the small part of LK is misclassified as pancreas. Compared to the label filling method, our approach exhibits a better ability to highlight tumor segmentation. It demonstrates improved tumor segmentation performance.

**Bad segmentation cases** In case-0067 of Fig. 4, the baseline has trouble in segmenting the IVC and aorta. And all three methods fail to segment the

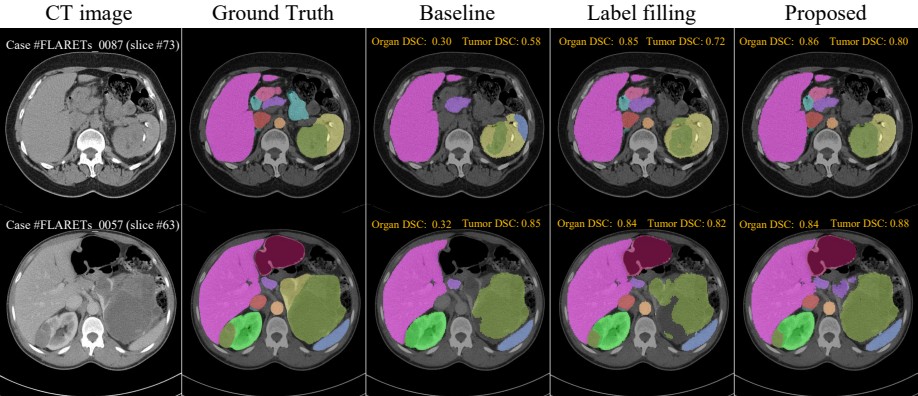

**Fig. 3.** Good segmentation cases from 50 validation set.

esophagus. It can be explained that the location of the esophagus makes all the methods confusing. In case-0095, as shown in Fig. 4, the baseline model does not segment the duodenum, IVC, gallbladder and aorta. The three methods misclassify the LK as the tumor. The duodenum and pancreas are similar in gray scale so the boundary of these organs is not clear in the predictive segmentation.

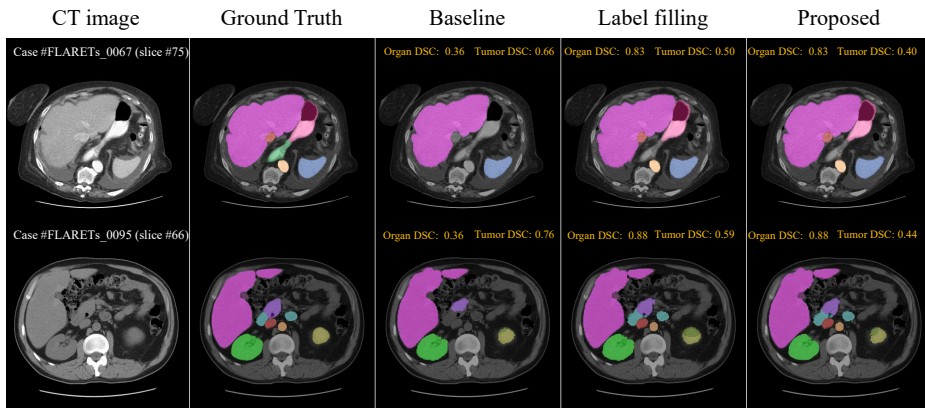

**Fig. 4.** Bad segmentation cases from 50 validation set.

### 4.3    Segmentation efficiency results on validation set

We have submitted our Docker container encapsulating our model to the official challenge. We have tested it on 20 cases, and the efficiency metrics were as

follows: an average execution time of 40.673 seconds, an average maximum GPU memory usage of 4499.8MB, and an average area under the CPU curve of 124628 seconds. There are 8 cases with efficiency as shown in Table 7.

**Table 7.** Quantitative evaluation of segmentation efficiency in terms of the running them and GPU memory consumption. Total GPU denotes the area under GPU Memory-Time curve. Evaluation GPU platform: NVIDIA QUADRO RTX5000 (16G).

| Case ID | Image Size | Running Time (s) | Max GPU (MB) | Total GPU (MB) |
|---------|------------|------------------|--------------|----------------|
| 0001 | (512, 512, 55) | 33.39 | 4088 | 75893 |
| 0051 | (512, 512, 100) | 43.83 | 4850 | 154144 |
| 0017 | (512, 512, 150) | 46.19 | 4938 | 161893 |
| 0019 | (512, 512, 215) | 41.23 | 4394 | 122667 |
| 0099 | (512, 512, 334) | 51.92 | 4686 | 155622 |
| 0063 | (512, 512, 448) | 53.18 | 4674 | 154248 |
| 0048 | (512, 512, 499) | 59.8 | 4658 | 175999 |
| 0029 | (512, 512, 554) | 75.38 | 5202 | 231308 |

### 4.4 Results on final testing set

The testing results from the docker of our solution were evaluated by the challenge officially on the Final Testing, and are shown in Table 5.

### 4.5 Limitation and future work

Upon reflecting on our study, it becomes evident that we encounter certain limitations in the following aspects.

**Calculation of Adaptive Weights:** The computation of adaptive weights did not take into consideration the issue of small organ volumes, resulting in a lack of differentiation in loss weights between small organs. Moreover, we find the phenomenon that the lower occurrence of label frequency resulted in a loss of segmentation accuracy, as evidenced by the fact that in our approach, while there was an improvement in tumor DSC, the mean DSC for organs experienced a slight decrease.

**Effect of Different Preprocessing Strategies:** Different preprocessing strategies were found to impact the contrast of the images. Future work may involve training on a fusion of images processed using various preprocessing methods.

**Frequency is not fully taken into account in modeling:** The frequency of each object is different in the dataset. Considering the frequency of each object would improve the segmentation performance of the model.

## 5    Conclusion

In order to tackle 'partial labeling/annotation problem', we develop an adaptive learning network, AdaptNet, to effectively segment multiple organs and tumors within partially labeled datasets from abdomen CT images. The quantitative and qualitative results show that AdaptNet can efficiently and flexibly learn multiple organ and tumor information from the partial labeling/annotation dataset, which is typically performed by multiple or multi-head networks. We conducted validation on a large-scale partially annotated dataset under MICCAI FLARE 2023 challenge and demonstrated that the proposed AdaptNet outperforms baseline segmentation methods across the 13 different organ and tumor segmentation tasks.

**Acknowledgements** The authors of this paper declare that the segmentation method they implemented for participation in the FLARE 2023 challenge has not used any pre-trained models nor additional datasets other than those provided by the organizers. The proposed solution is fully automatic without any manual intervention. We thank all the data owners for making the CT scans publicly available and CodaLab [22] for hosting the challenge platform. We also acknowledge Dr. Zheng Yunlin for kindly sharing the clinical knowledge and supporting some analysis for the segmentation results. This research was supported by the National Natural Science Foundation of China[No. 61972107]; Regional Innovation and Development Joint Fund of National Natural Science Foundation of China [No. U22A20345]. National Science Foundation for Young Scientists of China [No. 82202142]; China Postdoctoral Science Foundation [No. 2022M720857];Guangdong Provincial Key Laboratory of Artificial Intelligence in Medical Image Analysis and Application [No. 2022B1212010011]; High-level Hospital Construction Project [No. DFJHBF202105]; Open Project of Guangdong Provincial Key Laboratory of Artificial Intelligence in Medical Image Analysis and Application [No. 2022B1212010011].

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

**Table 8.** Checklist Table. Please fill out this checklist table in the answer column.

| Requirements | Answer |
| --- | --- |
| A meaningful title | Yes |
| The number of authors ($\leq 6$) | 6 |
| Author affiliations and ORCID | Yes |
| Corresponding author email is presented | Yes |
| Validation scores are presented in the abstract | Yes |
| Introduction includes at least three parts: background, related work, and motivation | Yes |
| A pipeline/network figure is provided | 2 |
| Pre-processing | 3-4 |
| Strategies to use the partial label | 4-7 |
| Strategies to use the unlabeled images. | 7 |
| Strategies to improve model inference | 6 |
| Post-processing | 6 |
| Dataset and evaluation metric section is presented | 7 |
| Environment setting table is provided | 2 |
| Training protocol table is provided | 3 |
| Ablation study | 11 |
| Efficiency evaluation results are provided | 7 |
| Visualized segmentation example is provided | 3, 4 |
| Limitation and future work are presented | Yes |
| Reference format is consistent. | Yes |