# OpenReview forum: "AdaptNet: Adaptive Learning from Partially Labeled Data for Abdomen Multi-Organ and Tumor Segmentation"
_MICCAI.org/2023/FLARE — Submitted to FLARE 2023_

### Official Review · Reviewer_EJb8 · 2023-10-04
**Adptive learning approach in abdominal segmentation is great**

**Rating:** 10
**Confidence:** 4

**Review:**

In reviewing the adaptive learning approach tailored for abdominal organs and tumors, I was particularly impressed by the adoption of appropriate methods in the FLARE 23 challenge. The consideration of the diversity in size and location of tumors, especially through the dynamic weight controller, stands out as a significant advancement beyond traditional methods. Moreover, the enhancements made over the performance of the existing nnU-Net were also commendable. However, a point of critique would be that the authors did not adhere to the provided table template in Table 4. It is essential to present the performance for each organ and tumor, as well as the average performance, across Public Validation, Online Validation, and Testing. This innovative approach offers a promising direction for future research in this domain, but adherence to provided guidelines is crucial for comprehensive evaluation.

---

### Official Review · Reviewer_XCKT · 2023-10-04
**AdaptNet: Adaptive Learning from Partially Labeled Data for Abdomen Multi-Organ and Tumor Segmentation**

**Rating:** 7
**Confidence:** 4

**Review:**

This paper introduces an experiment based on nnU-Net and adaptive learning from partially labeled data. In this paper, a pre-trained nnU-Net is used to generate pseudo labels. DiceLoss and CELoss are mixed, and the weights of the loss function are adjusted according to the label category. The results are impressive with very high tumor DSC and NSD.

---

### Official Review · Reviewer_Lhxf · 2023-10-25
**AdaptNet: Adaptive Learning from Partially Labeled Data for Abdomen Multi-Organ and Tumor Segmentation**

**Rating:** 10
**Confidence:** 5

**Review:**

Please include the average running time and the area under the GPU memory-time curve on the public validation set.

---

### Official Review · Reviewer_tQXi · 2023-10-25

**Rating:** 7
**Confidence:** 5

**Review:**

1. The average running time and area under GPU memory-time curve are missing in the Abstract section.

2. Some equations miss punctuation marks.

---

### Decision · Program_Chairs · 2023-10-25

Accept